# ARCS: Agentic Retrieval-Augmented Code Synthesis with Iterative Refinement

## Abstract

We present Agentic Retrieval-Augmented Code Synthesis (ARCS), a system that improves LLM-based code generation without fine-tuning. ARCS operates through a budgeted synthesize–execute–repair loop over a frozen model: it retrieves relevant code context before generation, proposes candidates, executes them against tests, and repairs based on execution feedback. This retrieval-before-generation design reduces hallucination and accelerates convergence. We formalize ARCS as a state-action process with provable guarantees on termination, monotonic improvement, and bounded cost. A tiered controller (Small/Medium/Large) trades latency for accuracy predictably. On HumanEval, ARCS achieves up to 87.2% pass@1 with Llama-3.1-405B, surpassing CodeAgent (82.3%) while using simpler control than tree-search methods. On TransCoder, it achieves $\geq 90\%$ accuracy on most translation pairs. On a LANL scientific corpus, it improves CodeBLEU by +0.115 over baseline RAG. ARCS provides a practical, reproducible approach to reliable code synthesis using existing LLM checkpoints.

## 1 Introduction

Large language models (LLMs) have achieved remarkable performance on code generation benchmarks. Despite these advances, state-of-the-art models often fail to consistently produce correct code, especially for complex tasks. Common issues include logical errors, incomplete functions, and reliance on hallucinated context. Current approaches frequently treat the model as a one-shot generator, forgoing iterative refinement strategies that leverage runtime feedback and knowledge retrieval.

We introduce *Agentic Retrieval-Augmented Code Synthesis (ARCS)*, a retrieval-before-generation framework that runs a budgeted *synthesize-execute-repair* loop over a **frozen** proposal model. In each round, ARCS (i) *retrieves* task-relevant project/API evidence to augment the prompt, (ii) *proposes* a candidate (or edit), (iii) *executes* it in a sandbox against tests to obtain *execution feedback*, and (iv) *repairs* the prompt by encoding that feedback for targeted revision. We formalize this procedure in Section 3.1 and prove guarantees in Section 3.3, instantiating it as a lightweight controller that guarantees termination under a fixed iteration budget while monotonically improving the best-so-far candidate within a run.

Our key contributions are:

- **Framework.** We cast iterative code synthesis as a budgeted state–action process with retrieval-augmented prompts and verification-in-the-loop updates over a frozen policy (Sec. 3).

- **System.** We instantiate the framework as an agentic RAG pipeline with optional planning, plan-conditioned retrieval, deterministic sandbox execution, and encoded repair (Sec. 3.2).

- **Tiers.** We expose Small/Medium/Large operational modes as strict projections of the full controller, yielding non-decreasing attainable success as capacity increases (Sec. 3.4).

- **Empirics.** On **HumanEval**, **TransCoder**, and a LANL scientific corpus, ARCS matches or exceeds strong prompting/agentic baselines while using fewer samples and comparable wall-clock time (Sec. 4).

## 2 RELATED WORK

A growing body of work augments code synthesis with external knowledge to bridge the gap between natural language intent and code Bhattarai et al. (2024a;b). Empirically, retrieving similar snippets or API usage examples can boost generation quality Yang et al. (2025). However, naive token-level retrieval may inject irrelevant or syntactically invalid code Zhang et al. (2023a). To mitigate this, methods constrain retrieval to task-relevant content—for example, kNN-TRANX narrows search to a task-specific datastore and uses syntax-aware matching to reduce noise Zhang et al. (2023b).

Complementary to retrieval, chain-of-thought (CoT) prompting asks models to produce intermediate reasoning (e.g., pseudocode) before emitting final code, often improving correctness. Structured CoT that guides planning with programming constructs can substantially raise pass@1 on standard benchmarks Li et al. (2023). That said, gains vary with model/backbone, and indiscriminate CoT can introduce spurious steps or errors Li et al. (2025); Liu et al. (2024).

Given the difficulty of one-shot correctness, many approaches interleave generation with execution and repair. AlphaCode demonstrated ex post execution checking by generating many candidates and filtering via tests Li et al. (2022). Newer agentic systems replace brute-force sampling with tool-using loops that retrieve documentation, write code, and run unit tests (e.g., CodeAgent) and report sizable pass-rate improvements on repository-level tasks Zhang et al. (2024). Search-based controllers further refine the loop: RethinkMCTS applies tree search over reasoning steps and incorporates fine-grained execution feedback, markedly improving pass@1 on HumanEval for certain backbones Brown & Green (2023). (Results across works typically use their original backbones and are not directly comparable.)

A separate line integrates execution feedback during decoding or training so the backbone internalizes correction behavior. Examples include running unit tests during decoding and/or continuing training with objectives that reward passing executions He et al. (2025); Zhu et al. (2024).

ARCS adopts a lightweight agentic loop with three design choices: (i) *retrieval before generation* (plan-conditioned when useful), (ii) *verification-in-the-loop* repair based solely on execution signals, and (iii) a *frozen* backbone with a budgeted controller that provides explicit termination and deterministic replay. Unlike large-sample filtering (e.g., AlphaCode) or complex tree search (e.g., RethinkMCTS), ARCS targets predictable accuracy–latency trade-offs under fixed iteration/retrieval budgets while retaining reproducibility.

## 3 THE ARCS FRAMEWORK

We present ARCS as a budgeted *synthesize-execute-repair* loop that transforms a natural-language specification into a correct program using a frozen proposal model. This section unifies the theoretical formulation with its practical instantiation: each component is described with a clear formal specification and its implementation details.

### 3.1 OVERVIEW AND CORE LOOP

In each round, the controller: (i) *retrieves* task-relevant code evidence to augment the prompt, (ii) *synthesizes* a candidate (or edit) from a frozen model $\pi_0$ given the composed prompt, (iii) *executes* it in a sandbox against a test suite to obtain verifiable feedback, and (iv) *repairs* the prompt by encoding that feedback (and, when enabled, updating a plan and evidence). The process halts when all tests pass or a fixed iteration budget $B$ is exhausted.

**Task and Notation.** Given a natural-language specification $q$, the goal is to produce a program $\hat{c}^*$ that satisfies a test suite $\mathcal{T} = \{(x_i, y_i)\}_{i=1}^m$. We assume $m \geq 1$ and treat any exception/timeout in the executor as a failure on the corresponding test. A candidate $\hat{c}$ is correct iff

$$\hat{c}(x_i) = y_i \quad \forall i \in \{1, \ldots, m\}. \tag{1}$$

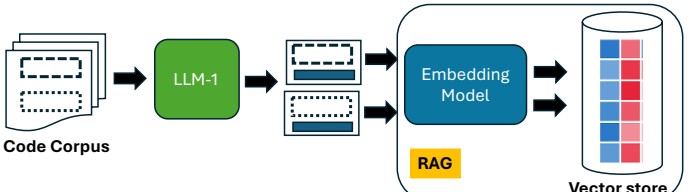

*Step 1: Extraction of core functional blocks and agentic labeling with LLM1 followed by embedding of labels for RAG registration*

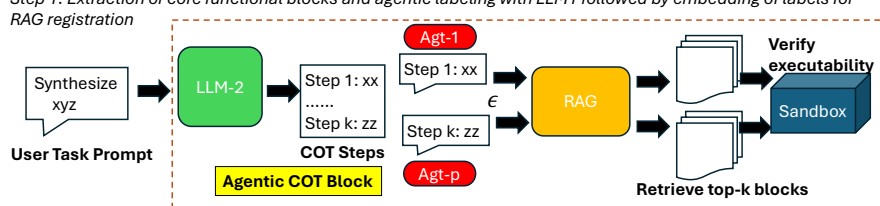

*Step 2: For a given user task prompt, generate COT steps with LLM-2, use each COT step as query to RAG to retrieve corresponding code blocks and validate.*

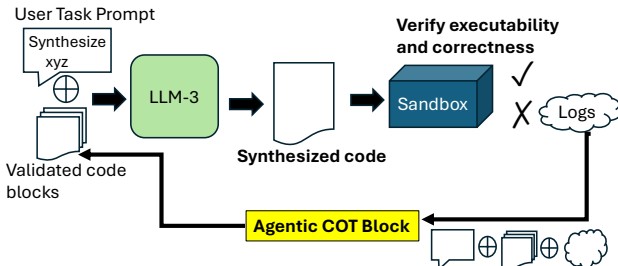

*Step 3: Feed the validated code blocks concatenated with User query to feed to LLM to synthesize final code and evaluate its correctness. If fails, parse output metadata including logs and errors and concatenate with query and code blocks to feed to Agentic COT block to generate new set of code blocks and repeat.*

Figure 1: High-level overview of ARCS. (1) Extract and embed code snippets with metadata. (2) Use CoT planning to issue plan-conditioned retrieval queries; render retrieved evidence into the prompt. (3) Propose code, execute in a sandbox against tests, and encode execution feedback to repair iteratively.

We assume a code corpus $\mathcal{C} = \{c_1, \ldots, c_N\}$ with metadata $\mathcal{M}_i$ (signatures, docstrings, comments). For retrieval, we use an embedding function $\Phi : \mathcal{X} \to \mathbb{R}^d$ and denote $\mathbf{e}_i = \Phi(\mathcal{M}_i)$; in our implementation $\Phi$ is `all-MiniLM-L6-v2`.

**Iterative Framework.** We model one iteration using the tuple $(\mathcal{S}, \mathcal{A}, \mathcal{P}, \mathcal{R})$:

- **State** $S_t = (q_t, s_t, \hat{c}_{t-1}, f_{<t}) \in \mathcal{S}$ where $q_t$ is the current prompt/context, $s_t$ is an optional CoT plan, $\hat{c}_{t-1}$ is the previous candidate (or $\varnothing$ at $t = 0$), and $f_{<t}$ are prior execution signals.

- **Action** $A_t \in \mathcal{A}$ is the proposed code (or edit). In our instantiation, $A_t \equiv \hat{c}_t$ sampled from $\pi_0$.

- **Transition** $\mathcal{P} : \mathcal{S} \times \mathcal{A} \to \mathcal{S}$ executes the proposal in a sandbox and returns $S_{t+1}$ with updated context and feedback.

- **Reward (optional)** $\mathcal{R} : \mathcal{S} \times \mathcal{A} \to \mathbb{R}$ measures improvement (e.g., the number of tests passed); used for selection/termination, not for weight updates.

The transition is

$$\hat{c}_t \sim \pi_0(\cdot \mid q'_t), \quad f_t = \mathcal{E}(\hat{c}_t, \mathcal{T}), \quad q_{t+1} = q_t \oplus \mathrm{Encode}(f_t), \tag{2}$$

yielding the next state:

$$S_{t+1} = (q_{t+1}, s_{t+1}, \hat{c}_t, f_{\leq t}), \quad \text{with } f_{\leq t} \text{ denoting the sequence } (f_{<t} \,\|\, f_t). \tag{3}$$

---

**Algorithm 1** ARCS (Large Tier)

---

**Require:** spec $q$, test suite $\mathcal{T}$, budgets $k$ (retrieval), $B$ (iterations)
**Ensure:** program $\hat{c}^*$ (or best-so-far $\hat{c}^\star$)
1: $q_0 \leftarrow q$; $s_0 \leftarrow \varnothing$; $\hat{c}_0 \leftarrow \varnothing$; $f_0 \leftarrow \varnothing$
2: **for** $t = 0 \rightarrow B-1$ **do**
3:     *(optional CoT planning)* $s_t \leftarrow P_\phi(q_t)$
4:     *(retrieval)* $\mathsf{R}_t \leftarrow \begin{cases} \mathsf{Ret}_k(q_t), & s_t = \varnothing \\ \mathrm{FilterRedundancy}\big(\bigcup_j \mathsf{Ret}_{k_j}(s_t^{(j)})\big), & \text{otherwise} \end{cases}$
5:     *(compose)* $q'_t \leftarrow q_t \oplus \bigoplus_{c \in \mathsf{R}_t} \Psi(c)$
6:     *(propose)* $\hat{c}_t \sim \pi_0(\cdot \mid q'_t)$
7:     *(execute)* $f_t \leftarrow \mathcal{E}(\hat{c}_t, \mathcal{T})$
8:     **if** $\hat{c}_t$ passes all tests **then**
9:         **return** $\hat{c}_t$
10:    **end if**
11:    *(repair)* $q_{t+1} \leftarrow q_t \oplus \mathrm{Encode}(f_t)$
12: **end for**
13: **return** $\arg\max_{\hat{c} \in \{\hat{c}_0, \ldots, \hat{c}_{B-1}\} \setminus \{\varnothing\}} \sum_{i=1}^m \mathbf{1}[\hat{c}(x_i) = y_i]$

---

The halting rule is:

$$\text{halt if } \frac{1}{m} \sum_{i=1}^m \mathbf{1}[\hat{c}_t(x_i) = y_i] = 1 \ \text{ or } \ t = B - 1. \tag{4}$$

We track the best-so-far candidate as

$$\hat{c}_t^\star = \arg \max_{\hat{c} \in \{\hat{c}_0, \ldots, \hat{c}_t\} \setminus \{\varnothing\}} \frac{1}{m} \sum_{i=1}^m \mathbf{1}[\hat{c}(x_i) = y_i]. \tag{5}$$

**Similarity and Filtering.** The similarity function used throughout is cosine similarity:

$$\mathrm{sim}(\phi_1, \phi_2) = \frac{\phi_1 \cdot \phi_2}{||\phi_1|| \cdot ||\phi_2||} \in [-1, 1]. \tag{6}$$

The redundancy filter (used in Eq. equation 11) removes near-duplicates:

$$\mathrm{FilterRedundancy}(\mathcal{R}) = \{c \in \mathcal{R} : \nexists c' \in \mathcal{R}, \ \mathrm{sim}(\mathbf{e}(c), \mathbf{e}(c')) > \delta \wedge \mathrm{index}(c') < \mathrm{index}(c)\}, \tag{7}$$

where $\mathbf{e}(c) := \Phi(\mathcal{M}(c))$ and $\mathrm{index}(\cdot)$ denotes the fixed corpus ordering; we use $\delta = 0.85$.

**Design Guarantees.** Termination is guaranteed after at most $B$ iterations by Eq. equation 4; the best-so-far score in Eq. equation 5 is non-decreasing in $t$ since the maximization set grows with $t$; and exact replay is enabled by fixing seeds, freezing the index snapshot, and using deterministic tie-breaking in retrieval (below).

## 3.2 Core Components: Theory and Implementation

**Planner $P_\phi$.** *Specification:* An optional CoT planner emits a plan $s_t = (s_t^{(1)}, \ldots, s_t^{(K)})$, where each $s_t^{(j)}$ is a subgoal at iteration $t$. When unused, $s_t = \varnothing$ (i.e., $K = 0$). *Implementation:* A lightweight prompt produces (i) a typed I/O contract, (ii) a pseudocode sketch, and (iii) up to $K \leq 4$ named subgoals. We canonicalize symbols, deduplicate, and drop conflicts. A gate disables planning when $|q_t| < 120$ tokens. Subgoal names are used directly as retrieval subqueries.

**Retriever $R_\psi$.** *Specification:* Given $q_t$ (or subquery), embed $\phi_t = \Phi(q_t)$ and compute scores $\sigma_i = \mathrm{sim}(\phi_t, \mathbf{e}_i)$. Define the ordered index set

$$I_k(q_t) = \arg \mathrm{topK}_{i \in \{1, \ldots, N\}} \sigma_i \quad \text{(ties broken deterministically by ascending index } i\text{)}, \tag{8}$$

and return

$$\mathsf{Ret}_k(q_t) = (c_{i_1}, \ldots, c_{i_k}) \ \text{ with } \ (i_1, \ldots, i_k) = I_k(q_t). \tag{9}$$

When $s_t \neq \varnothing$, define subqueries $q_t^{(j)} := s_t^{(j)}$ and retrieve per component with budgets $k_j$:

$$\mathsf{Ret}_{k_j}(q_t^{(j)}) = (c_{i_1^{(j)}}, \ldots, c_{i_{k_j}^{(j)}}), \tag{10}$$

then aggregate and deduplicate

$$\mathsf{Ret}(q_t) = \text{FilterRedundancy}\left(\bigcup_{j=1}^{K} \mathsf{Ret}_{k_j}(q_t^{(j)})\right). \tag{11}$$

*Implementation:* FAISS ANN index with cosine similarity over $\mathbf{e}_i$. Query with $q_t$ when $s_t = \varnothing$ or with $\{s_t^{(j)}\}$ otherwise, aggregate per Eq. equation 11. Near-duplicates are removed via cosine threshold ($\delta = 0.85$); deprecated/conflicting APIs are filtered via a denylist. Snapshots are frozen for experiments.

**Context Renderer and Proposer.** *Specification:* Render retrieved content via $\Psi : \mathcal{C} \rightarrow \mathcal{X}$:

$$\Psi(c) = \text{Format}(\text{signature}(c), \text{doc}(c), \text{code}(c)), \tag{12}$$

and compose the enriched prompt as

$$q_t' = q_t \oplus \bigoplus_{c \in \mathsf{R}_t} \Psi(c), \tag{13}$$

where $\mathsf{R}_t$ is produced either by single-query Top-$k$ (Eq. equation 9) or by plan-conditioned retrieval with deduplication (Eq. equation 11). The frozen proposal model then generates

$$\hat{c}_t \sim \pi_0(\cdot \mid q_t'). \tag{14}$$

*Implementation:* Frozen Llama-3.x checkpoints (*Meta-Llama-3.1-70B/3.3-70B/3.1-405B Instruct*), nucleus sampling (temperature 0.7, top-$p = 0.95$, $L_{\max} = 512$, seed 42). Encoders: $\text{Enc}_{\text{env}}$ (runtime constraints), $\text{Enc}_{\text{plan}}$ (subtasks), $\text{Enc}_{\text{evid}}$ (retrieved code), $\text{Enc}_{\text{inv}}$ (interface invariants).

**Executor $\mathcal{E}$.** *Specification:* Execute $\hat{c}_t$ w.r.t. $\mathcal{T}$:

$$f_t = \mathcal{E}(\hat{c}_t, \mathcal{T}). \tag{15}$$

*Implementation:* Docker sandbox (no network; read-only FS except a temp workspace). Per-test caps: 10s wall-clock, 4GB memory. We capture exit codes, stdout/stderr, and exception metadata. $\text{Enc}_{\text{fb}}$ serializes pass/fail vectors and truncated stacks.

**Repair and Refresh.** *Specification:* Append feedback and optionally re-plan:

$$q_{t+1} = q_t \oplus \text{Encode}(f_t), \qquad s_{t+1} = P_\phi(q_{t+1}; s_t, f_t). \tag{16}$$

Refresh evidence under the updated query:

$$\text{Refresh}(q_{t+1}, s_{t+1}) \triangleq \begin{cases} \mathsf{Ret}_k(q_{t+1}) & \text{if } s_{t+1} = \varnothing, \\ \text{FilterRedundancy}\left(\bigcup_{j=1}^{K} \mathsf{Ret}_{k_j}(s_{t+1}^{(j)})\right) & \text{otherwise.} \end{cases} \tag{17}$$

*Implementation:* $\text{Enc}_{\text{fb}}$ summarizes only *observed* failures (exception type, failing inputs, truncated stack). On context pressure, apply stable truncation: keep $\text{Enc}_{\text{inv}}$ and the most recent failing trace; drop lowest-similarity evidence first; collapse $\text{Enc}_{\text{plan}}$ to headers.

**Computational considerations.** Using an ANN index, retrieval per round typically costs sublinear time in $N$ (e.g., $O(\log N)$ or $O(1)$ average with LSH-style structures), so one iteration is dominated by retrieval + prompting/generation + execution. Concretely, each round incurs $O(\text{ANN\_probe}(k, N))$ for retrieval, $O(|q_t'|)$ for generation, and $O(m \cdot T_{\text{exec}})$ for execution; total cost scales linearly with the budget $B$ (see Prop. 3).

## 3.3 THEORETICAL PROPERTIES AND GUARANTEES

**Proposition 1** (Monotonic Improvement). *Under best-so-far tracking (Eq. equation 5), the success metric is non-decreasing:*

$$\forall t : \quad \frac{1}{m} \sum_{i=1}^{m} \mathbf{1}[\hat{c}_{t+1}^{\star}(x_i) = y_i] \;\geq\; \frac{1}{m} \sum_{i=1}^{m} \mathbf{1}[\hat{c}_t^{\star}(x_i) = y_i].$$

*Proof.* $\{\hat{c}_0, \ldots, \hat{c}_{t+1}\} \setminus \{\varnothing\} \supseteq \{\hat{c}_0, \ldots, \hat{c}_t\} \setminus \{\varnothing\}$, so the maximized value cannot decrease. $\square$

**Proposition 2** (Bounded Termination). *Algorithm 1 terminates in at most $B$ iterations.*

*Proof.* The loop runs for $t = 0, \ldots, B - 1$ with an early return when all tests pass (Eq. equation 4); hence termination is guaranteed after at most $B$ iterations. $\square$

**Proposition 3** (Cost Bound). *Let $C_{\mathrm{retr}}(k)$ bound the per-round retrieval cost (including ANN probes and formatting of the top-$k$ items), $C_{\mathrm{in}}(L)$ and $C_{\mathrm{out}}(L)$ bound the LLM's input and output token costs for length $L$, and $C_{\mathrm{exec}}$ bound the per-test execution cost in $\mathcal{E}$. If $|q_t'| \leq L_{\max}$ and $\hat{c}_t$ has at most $L_{\max}$ output tokens each round, then*

$$C_{\mathrm{total}} \;\leq\; B \cdot \big( C_{\mathrm{retr}}(k) + C_{\mathrm{in}}(L_{\max}) + C_{\mathrm{out}}(L_{\max}) + m \cdot C_{\mathrm{exec}} \big). \tag{18}$$

*Proof.* Each round incurs retrieval, LLM input/output, and up to $m$ test executions. Upper-bounding each term and summing over at most $B$ rounds yields the result. $\square$

We do not claim general convergence guarantees for arbitrary tasks. Intuitively, feedback encoding induces an *effective* policy $\pi_t(c \mid q) = \pi_0(c \mid q \oplus f_{<t})$; when tests provide informative signals and the target lies within $\pi_0$'s support, additional rounds can improve success.

**Lemma 1** (Tier Monotonicity). *Fix the backbone $\pi_0$, corpus/index, and benchmark distribution. Define the attainable success probability of a tier as the supremum over its controller parameters and internal randomness. Then $\mathcal{P}_S \leq \mathcal{P}_M \leq \mathcal{P}_L$.*

*Proof.* Small is the subset of controllers with $(k = 0, s_t = \varnothing, B = 1)$. Medium enlarges this set to allow $s_t \neq \varnothing$ but keeps $(k = 0, B = 1)$. Large further enlarges to $k > 0$ and $B > 1$. Since each controller class is a subset of the next, the supremum success over a subset is at most that over its superset. $\square$

## 3.4 TIER DESIGN AND OPERATIONAL MODES

We expose a tiered interface to provide predictable accuracy-latency trade-offs and enable clean ablations. Each tier is a strict projection of the full controller, preserving the guarantees above:

- **Small (one-shot):** No planning or retrieval, one round—$(k = 0, s_t = \varnothing, B = 1)$. Compose, propose, test.

- **Medium (structured single-shot):** Planning enabled, retrieval disabled, one round—$(k = 0, s_t \neq \varnothing, B = 1)$. Exploits decomposition when specifications are complete.

- **Large (retrieval+repair):** Planning, plan-conditioned retrieval with refresh, multi-round repair—$(k > 0, s_t \neq \varnothing, B > 1)$.

Default parameters: $k = 10$ (retrieval budget), $B = 5$ (iteration budget), $K \leq 4$ (max subgoals). By Lemma 1, these tiers form a hierarchy where each strictly contains the capabilities of the previous, ensuring users can predictably trade compute for accuracy.

Table 1: HumanEval pass@1. External baselines shown with original backbones; ARCS uses frozen Llama-3.1-70B.

| Method | Backbone LLM | pass@1 |
|---|---|---|
| GPT-3.5-Turbo (Baseline) | GPT-3.5-Turbo | 72.6% |
| CodeAgent | GPT-3.5-Turbo | 82.3% |
| RethinkMCTS | GPT-3.5-Turbo | 89.0% |
| ARCS (Small) | Llama-3.1-70B | 79.9% |
| ARCS (Medium) | Llama-3.1-70B | 76.8% |
| ARCS (Large) | Llama-3.1-70B | 83.5% |

## 3.5 SYSTEM IMPLEMENTATION AND REPRODUCIBILITY

The ARCS system runs on the SambaNova Systems platform with real-time LLM API access. The framework is modular Python code that queries live code repositories for retrieval; for experiments we freeze snapshots by embedding metadata offline and indexing it to avoid runtime drift. For deterministic performance, we employ versioned LLM checkpoints and FAISS indices, fixed seeds and decoding hyperparameters, and complete logging of $(q_t, s_t, q'_t, \hat{c}_t, f_t)$ per round with index snapshots.

## 4 EXPERIMENTS AND RESULTS

We evaluate ARCS on standard code-generation and code-translation benchmarks and on a domain-specific scientific corpus. Our evaluation probes: (i) accuracy under a fixed compute budget as formalized by the controller loop (Sec. 3.1), (ii) the contribution of retrieval, planning, and verification through ablations grounded in Eqs. equation 13–equation 16, and (iii) robustness on real scientific libraries. Unless noted, $\pi_0$ is a frozen Llama-3.x variant; decoding uses fixed temperature/top-$p$ and a fixed random seed; the executor $\mathcal{E}$ is deterministic under fixed time/memory caps. To avoid index contamination, the retrieval corpus excludes benchmark references and is filtered for near-duplicates via cosine thresholds and token-hash screening.

### 4.1 BENCHMARKS AND METRICS

**HumanEval** Chen et al. (2021) contains 164 Python problems with hidden unit tests. We report pass@k, emphasizing pass@1 (the final candidate after the ARCS loop). **TransCoder** Rozière et al. (2020) evaluates translation among Python, Java, and C++; we report translation accuracy as the fraction of items whose translated code passes all tests. **LANL scientific corpus:** we select four repositories from `github.com/lanl` (`pyDNMFk`, `pyDNTNk`, `AdversarialTensors`, `EPBD_BERT`) and construct prompts from READMEs (documentation-based) and source code (code-based). We evaluate with CodeBLEU Zhang et al. (2020).

**Protocol and budgets.** Unless noted otherwise, we use the Large tier defaults ($k$=10, $B$=5, $K \leq 4$) and a single evaluation run per setting; pass@1 reflects the candidate returned by Eq. equation 4. Comparisons to external baselines are reported with their original backbones (see table captions); scores are indicative rather than strictly comparable across different backbones.

### 4.2 MAIN RESULTS ON HUMANEVAL

Table 1 shows that ARCS (Large) achieves competitive pass@1 with substantially simpler control than tree-search methods. While RethinkMCTS attains a higher score, it relies on heavier exploration. ARCS surpasses CodeAgent and a one-shot GPT-3.5 baseline while adhering to bounded budgets (Eq. equation 18). The pattern matches the framework: retrieval-before-generation focuses proposals on on-target regions (Eq. equation 13), and verification-guided repair converts execution signals into directed edits (Eq. equation 16), yielding monotone improvement in the best-so-far score (Eq. equation 5) within the halting rule.

Table 2: Ablation on HumanEval (pass@1). Rows add components to a one-shot baseline (frozen $\pi_0$, no retrieval, no CoT, no feedback).

| Configuration | pass@1 |
|---|---|
| Baseline (no RAG, no CoT, no Feedback) | 72.6% |
| + Retrieval only | 75.2% (+2.6 pp) |
| + CoT only | 76.1% (+3.5 pp) |
| + Execution feedback only | 74.8% (+2.2 pp) |
| + Retrieval + CoT | 78.4% (+5.8 pp) |
| + Retrieval + Feedback | 77.3% (+4.7 pp) |
| + CoT + Feedback | 79.1% (+6.5 pp) |
| ARCS (Full) | 83.5% (+10.9 pp) |

Table 3: HumanEval pass@1 across ARCS tiers and Llama-3.x backbones.

| Backbone | Small | Medium | Large |
|---|---|---|---|
| Llama-3.1-70B-Instruct | 79.9% | 76.8% | 83.5% |
| Llama-3.3-70B-Instruct | 81.2% | 78.1% | 85.1% |
| Llama-3.1-405B-Instruct | 84.7% | 81.3% | 87.2% |

### 4.3 ABLATION: RETRIEVAL, PLANNING, AND VERIFICATION

Each component in Table 2 yields measurable gains; their combination is super-additive. Retrieval implements Eq. equation 13, reducing off-target proposals; planning decomposes queries (multi-component retrieval, Eqs. equation 10–equation 11); and execution feedback implements Eq. equation 16, injecting ground truth rather than speculative heuristics. The full loop realizes the intended synergy.

### 4.4 HUMANEVAL ACROSS BACKBONES AND TIERS

Scaling the backbone consistently improves accuracy (Table 3). Large dominates on problems requiring nontrivial repair, as expected from the iterative loop; Medium can underperform Small on simple functions due to over-structuring when verification is absent. This matches the design: Medium introduces structure without the corrective execution signal of Eq. equation 16, whereas Large leverages verification to align with Eq. equation 1. Compute trade-offs follow Eq. equation 18: Large incurs multiple rounds but reduces failed candidates per success; empirically, $B \in [2, 5]$ captures most gains.

### 4.5 TRANSCODER: CROSS-LANGUAGE TRANSLATION

Table 4 shows strong accuracy across directions. Medium often matches or slightly exceeds Large, consistent with translation benefiting from structured decomposition without repeated verification. The most challenging direction is Python→C++ due to the paradigm shift; Java→C++ is easier given syntactic and semantic alignment. When exact outputs are required, Large's verification corrects occasional semantic mismatches in subsequent rounds.

### 4.6 LANL SCIENTIFIC CORPUS: CODEBLEU

On realistic scientific code, ARCS substantially outperforms a one-shot RAG baseline (Table 5). Gains in weighted n-grams and syntax indicate that retrieval-before-generation (Eq. equation 13) guides the model toward project-specific idioms/APIs, while iterative repair (Eq. equation 16) increases structural and semantic fidelity beyond token overlap.

Table 4: TransCoder accuracy (%). Each cell shows Small/Medium/Large for ARCS.

| Backbone | C++→Py | Py→C++ | Java→Py | Java→C++ |
|---|---|---|---|---|
| Llama-3.1-70B | 84.8/88.8/89.1 | 83.3/84.1/84.9 | 86.5/89.5/88.9 | 95.5/93.9/94.5 |
| Llama-3.3-70B | 86.3/89.5/88.4 | 85.9/85.1/86.9 | 89.5/91.2/90.5 | 94.5/94.8/94.3 |
| Llama-3.1-405B | 90.5/91.6/91.5 | 86.0/89.5/87.0 | 90.6/91.7/91.6 | 96.4/95.5/96.4 |
| TransCoder Rozière et al. (2020) | 67.2 | 57.3 | 68.7 | 91.6 |

Table 5: CodeBLEU on the LANL corpus: one-shot RAG baseline vs. ARCS (Large).

| Metric | Basic RAG | ARCS (Large) |
|---|---|---|
| Overall CodeBLEU | 0.289 | 0.404 (+0.115) |
| N-gram Match | 0.105 | 0.232 (+0.127) |
| Weighted N-gram Match | 0.131 | 0.294 (+0.163) |
| Syntax Match | 0.527 | 0.640 (+0.113) |
| Dataflow Match | 0.227 | 0.284 (+0.057) |

## 4.7 DISCUSSION AND LIMITATIONS

The empirical trends align with ARCS's design. *Retrieval-before-generation* focuses proposals on on-target regions, and *verification-in-the-loop* injects a corrective signal that translates into improved best-so-far scores via the repair step and the monotonicity property established in Proposition. Tier behavior follows the projections argument: Large subsumes Small and Medium and excels on problems that require repair; Medium's structured decomposition is advantageous for translation, but can over-structure trivial functions when verification is absent (consistent with non-decreasing *attainable* success probabilities in Lemma 1, even if a specific fixed configuration of Medium underperforms Small on easy items).

Regarding limitations,external baselines use different backbones; we report their original numbers for context, not as strictly controlled head-to-head comparisons. Despite filtering (cosine thresholding and token-hash screening), residual near-duplicates may persist and inflate retrieval utility. We mitigate this by freezing the index snapshot and deduplicating at inference (Sec. 3.2). Unit tests that require literal equality can penalize semantically acceptable variants; conversely, underspecified tests may overestimate correctness.

ARCS's correctness is test-suite relative; stronger oracles (metamorphic tests, property-based testing, or lightweight SMT checks) could improve robustness. Tier and budget selection is manual; a learned difficulty estimator that adapts $(B, k)$ while preserving bounded cost (Proposition 3) is a natural extension. Finally, we evaluated mostly function-level synthesis and TransCoder-style translation; repository-level tasks, richer languages/runtimes, and integration with formal tools are promising avenues.

## 5 CONCLUSION

We introduced ARCS, an agentic framework for retrieval-augmented code synthesis that integrates structured retrieval, optional planning, and execution-verified refinement over frozen LLMs. By casting synthesis as a budget-constrained state–action search (Sec. 3), ARCS provides bounded cost (Prop. 3), monotone improvement (Prop. 1), and practical reproducibility. Empirically, ARCS achieves up to 87.2% pass@1 on HumanEval, high accuracy on TransCoder, and +0.115 CodeBLEU improvement on scientific code-validating that retrieval-before-generation and verification-guided repair yield reliable gains without fine-tuning. Implementation details are provided to enable exact replay and extension.

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
