# OpenReview forum: "ARCS: Agentic Retrieval-Augmented Code Synthesis with Iterative Refinement"
_ICLR.cc/2026/Conference — ICLR 2026 Conference Withdrawn Submission_

### Official Review · Reviewer_MLDf · 2025-10-29

**Soundness:** 2
**Presentation:** 2
**Contribution:** 2
**Rating:** 2
**Confidence:** 4

**Summary:**

This paper studies agentic retrieval-augmented code generation, introducing a synthesize–execute–repair loop in which the system retrieves relevant code contexts, proposes candidate solutions, executes them, and repairs errors iteratively. The framework is instantiated using LLM-based agents.

**Strengths:**

- Timely topic: Agentic code generation is a trending and potentially impactful direction in AI4SE.
- The ablation study is informative and contributes useful insights about component contributions.

**Weaknesses:**

1. Technical originality and clarity

The proposed solution is technically straightforward. While agentic code generation and RAG-based code generation are each useful, their combination in this paper feels somewhat contrived rather than synergistic. The synthesis of these ideas does not lead to substantial new capabilities or theoretical insights.

Much of the technical content lacks depth. For example, Section 3.3 provides only trivial observations (e.g., termination due to an explicit upper bound), and it is unclear why such results need to be emphasized.

The reinforcement learning–like formulation (i.e., state–action process) is presented but not operational; it plays no meaningful role in the methodology or experiments.

2. Experimental evaluation

While using HumanEval is natural, the choice to include cross-language translation as an additional evaluation setting is poorly motivated. The use of CodeBLEU for the LANL dataset does not clearly demonstrate the advantages of the ARCS framework.

The observed improvements are modest—ARCS (87.2%) vs. CodeAgent (82.3%) on HumanEval—and are not particularly convincing, especially given the many uncontrolled factors that may influence the results.

The reliability of the results depends heavily on the quality and completeness of the test suites, which are not fully discussed.

3. Presentation quality

The paper is generally poorly written. The formalization in Section 3.2 complicates simple ideas rather than clarifying them.

Several technical terms are introduced without adequate explanation or citation — for example, “FAISS ANN index.” Such omissions unfortunately compromise readability.

**Questions:**

1. Are there any genuine innovations within each component of the proposed framework, or are these primarily adaptations of existing techniques?
2. What exactly does “frozen model” mean in this context? Does it refer to a model whose parameters are fixed during all stages (retrieval, synthesis, repair), or are there components that remain trainable?
3. What role does the state–action pair (i.e., the reinforcement learning formulation) play in the current implementation? It is introduced in the text, but its function and impact on the overall framework are unclear.

Technical:
Eq(2) what does \oplus exactly stand for?
Eq(5) what does \emptyset exactly stand for?
Line 072: ex post execution checking ? (a typo?)
Line 501 and 526: double reference entries.

---

### Official Review · Reviewer_D1WX · 2025-10-30

**Soundness:** 3
**Presentation:** 3
**Contribution:** 3
**Rating:** 6
**Confidence:** 5

**Summary:**

The paper presents Agentic Retrieval-Augmented Code Synthesis (ARCS) for improving LLM-based code generation through a synthesize–execute–repair loop. The authors show that their retrieval-before-generation design reduces hallucination and accelerates convergence, as well as improves accuracy. The authors approach of using chain of thought planning to decompose tasks into subgoals, and then retrieve code snippets for each subgoal separately is conceptually very sound. While many methods propose trading more compute for high performance, this paper includes provable bounds on termination, monotonic improvement, and cost. Lastly, the approach can keep the model frozen, and hence is more easily adaptable. The results show that on HumanEval, TransCoder, and a LANL scientific corpus, the ARCS technique matches or exceeds strong prompting.

**Strengths:**

- The authors present a clear formalization of the state-action-transition-reward iterative framework for their approach, as well as design guarantees based on inputs like the budget.
- Figure 1, the system architecture is clear presentation of the approach.
- The theoretical guarantees are relatively straight-forward, however the inclusion in the paper is a nice addition.
- The approach is novel, particularly by pairing planning with conditioned treival and the verification-in-the-loop. The verification and feedback has both conceptual and in the ablation shows strong emperical advantages without (presumably) adding signficant computational overhead.
- The experimental setup presented is thorough. Specifically, they include ablation studies looking at individual components in the system, include multiple benchmarks (TransCode, LANL, and HumanEval) for assessing performance under a variety of contexts, and compare against 3 baselines. Additionally, in Table 5 they compare against a basic RAG approach and consistently show superior performance.

**Weaknesses:**

- It seems like the Medium ARCS tier often underperforms small. It's unclear what the value in the Medium approach would be or when it is a good choice.
- In Table 1 why is the baseline model GPT-3.5-Turbo and not the same model as used in the ARCS backbone — this feels slightly unfair.
- Lacks statistical significance testing. Since the approach relies on frozen models, it is reasonable to run multiple trials.
- More could be added to the Related Work to distinguish the novelty of ARCS. Specifically, how the execution feedback differs from RethinkMCTS and how loop for ARCS differs from CodeAgent.
- No analysis of the retrieval provided. It's unclear how much the system depends on accurate retrieval and the impact of the choice of budget k on performance.
- Qualitative analysis to show failures and more insights into how the agentic loops like would be very valuable to readers.
- Empirical calculations of the costs to use the approach (e.g., latency or wall clock time) compared with a baseline like RAG would be helpful.

**Questions:**

- Why does the approach rely on RAG? Since you are already relying on a sandbox for execution, could the agents execute queries based on their plan. For example, if the agent needs a specific function, then it could grep within a python REPL over the code corpus. This type of approach would eliminate the need for a retriever and vector database.

---

### Official Review · Reviewer_ysBL · 2025-11-01

**Soundness:** 1
**Presentation:** 2
**Contribution:** 2
**Rating:** 0
**Confidence:** 4

**Summary:**

The paper uses RAG to retrieve code snippets to generate candidate solutions and uses execution feedback to adjust the solution. They test their approach on 3 coding benchmarks. They also claim to contribute theoretical properties of their algorithm.

**Strengths:**

-The paper is generally well written and easy to follow.

-To my knowledge, this is the first paper that I have seen that does RAG with code snippets so the application of retrieval is novel

**Weaknesses:**

-Table 1 shows higher performance with RethinkMCTS and the downside the authors mention that in Line 373 that it relies on “heavier exploration” does not satisfy me. That would suggest that there are situations where more exploration or better exploration strategies than what ARCS does is needed.


-The theoretical section does not contribute anything in my opinion. All propositions and lemmas are trivial and just based on definition. For proposition 1, best-so-far tracking by definition is monotonic. For proposition 2, if you set a loop for at most B iterations, then it will undoubtedly be bounded. For proposition 3, if you defined upper bounds for each portion of the cost then Eq 18 is apparent. Proposition 4 is saying that if you have more controllers to choose from, then the probability of having the optimal one is higher. These are not insightful theoretical properties.

**Questions:**

Suggestion:
For each portion of the method, you can remove the work “Specification”

---

### Official Review · Reviewer_wbjf · 2025-11-02

**Soundness:** 2
**Presentation:** 1
**Contribution:** 2
**Rating:** 2
**Confidence:** 3

**Summary:**

The paper proposes a method called Agentic Retrieval-Augmented Code Synthesis (ARCS) for iteratively engineering prompts fed into a fixed, backbone LLM for generating code to solve benchmark coding tasks. The proposed method has four main steps: (i) use Chain-of-Though (CoT) techniques to produce an LLM-generated description of a high-level idea for the solution approach based on the current prompt and concatenate this description with the prompt; (ii) retrieve relevant code examples from an existing code database, eliminate redundancies, and concatenate these examples with the prompt; (iii) generate code using the backbone LLM and current prompt; (iv) run unit tests, collect feedback, and incorporate this feedback into the prompt. Formal description of steps (i)-(iv) is provided. The proposed method with various versions of Llama-3.1 as backbone is experimentally compared with existing baselines using non-Llama backbones. Ablations illustrating the effect of the various components of (i)-(iv) are provided.

**Strengths:**

The proposed method is an intuitive and natural way to improve on existing retrieval-augmented methods for code generation. The experimental results indicate that the method is consistently able to solve challenging code generation tasks.

**Weaknesses:**

The paper has the following weaknesses:
1. The paper suffers from clarity and readability issues. It is unclear from the introduction and related works why previous methods for retrieval-augmented code generation fail and how the ARCS framework overcomes these issues. Throughout the paper the writing is very terse and full of undefined jargon, and several important claims lack justification or explanation (see **Questions** for examples). In addition, there are several points throughout the main body where important concepts and notation are inadequately defined or not defined altogether (see **Questions** for examples).
2. Several key components of the proposed method are not described in sufficient detail for reproducibility and are not situated clearly (with appropriate references and discussion) in the relevant literature. In particular, the descriptions provided in the *Implementation* sections in Section 3.2 of how the core components of the method were implemented are insufficient for reproducibility, and no code is provided to mitigate this issue. In addition, it is unclear from the description of the method what existing methods (for CoT, encoding, redundancy filtering, etc., with appropriate references) are leveraged within each component.
3. The experimental results do not compare ARCS against existing baselines with the same backbone LLM. This seriously undermines the meaningfulness of the experimental comparisons, since superior performance of ARCS against baseline methods could be due to the underlying backbone of ARCS (Llama-3.1-*) being better suited to the task at hand than the backbone used by the baseline, instead of inherent superiority of ARCS. From the description of ARCS, it is unclear why identical backbones were not used, seriously weakening the experimental contribution of the paper.

**Questions:**

1. What are the specific drawbacks of existing methods that ARCS addresses?
2. What are $x_i, y_i$ near lines 105-106?
3. What is $\mathcal{X}$ near lines 142-143?
4. What is all-MiniLM-L6-v2, lines 143-144?
5. What are $\mathcal{E}$ and Encode in equation (2)?
6. What does $(f_{<t} \lVert f_t)$ mean in equation (3)?
7. What are $\pi_1, \pi_2, \mathcal{R}$ in the **Similarity and Filtering** section on page 4?
8. What are Format(), signature(), doc(), and code() in equation (12)?
9. How are $Enc_{env}, Enc_{plan}, Enc_{evid}, Enc_{inv}$ defined on lines 244-245?
10. What is $P_{\phi}$ in equation (16)?
11. What do the fragments "On context pressure, apply stable trunction" and "collapse $Enc_{plan}$ to headers" mean in lines 263-264?
12. What are the definitions of $\mathcal{P}_{S}, \mathcal{P}_{M}, \mathcal{P}_{L}$ in Lemma 1?
13. Does the fact that ARCS (Small) outperforms ARCS (Medium) on pass@1 contradict Lemma 1?
14. What does the statement around lines 368-369 that "scores are indicate rather than strictly comparable across different backbones" mean?
15. What does it mean around lines 372-373that "ARCS (Large) achieves competitive pass@1 with substantially simpler control than tree-search methods"? Specifically, what does "simpler control" mean in this context?
16. Is there a concrete, quantitative meaning to the claim that "RethinkMCTS...relies on heavier exploration" around lines 373-374?
17. What does it mean at the end of page 7 that "the pattern matches the framework"? What does "the pattern"  refer to here?
18. Does the statement at the beginning of Section 4.5 that "Medium often matches or slightly exceeds Large, consistent with translation benifiting [sic] from structured decomposition without repeated verification" contradict Lemma 1?
19. What does it mean around lines 421-422 that "The most challenging direction is Python $\rightarrow$ C++ due to the paradigm shift"? What is "the paradigm shift"?

---

### Note · Authors · 2025-11-13

I have read and agree with the venue's withdrawal policy on behalf of myself and my co-authors.